**Data Availability Statement:** The dataset generated and analyzed during the current study

# Comparing the effect of childbirth preparation courses delivered both in-person and via social media on pregnancy experience, fear of childbirth, birth preference and mode of birth in pregnant Iranian women: A quasi-experimental study

**Seyedeh Robab Mousavi**[1], **Leila Amiri-Farahani**[2]*, **Shima Haghani**[3], **Sally Pezaro**[4]

1 Department of Reproductive Health and Midwifery, Faculty of Nursing and Midwifery, Iran University of Medical Sciences, Tehran, Iran, 2 Department of Reproductive Health and Midwifery, Nursing Care Research Center (NCRC), School of Nursing and Midwifery, Iran University of Medical Sciences, Tehran, Iran, 3 Department of Biostatistics, Nursing Care Research Center, Iran University of Medical Sciences, Tehran, Iran, 4 The Centre for Healthcare Research, Coventry University, Coventry, United Kingdom

* l.amirifarahani@gmail.com

## Abstract

### Background and aim

Rates of cesarean section in Iran are unnecessarily high largely due to fear of childbirth (FOC), yet this may be reduced through education. Iranian women are keen to obtain information about pregnancy and birth online though sources may not be reliable. Consequently, the present study aimed to compare the effect of childbirth preparation courses delivered both online via the social media platform 'Telegram' and in-person on pregnancy experience, FOC, birth preference, and mode of birth.

### Methods

This quasi-experimental study included 165 primiparous pregnant women referred to the prenatal clinic in Tehran, Iran. Convenience sampling was used to recruit participants, who were subsequently divided into three groups; (A) social media-based educational intervention (n = 53); (B) in-person educational intervention (n = 52), and (C) a control group who received no prenatal education (n = 50). During the 18th and 20th weeks of pregnancy, demographic questions along with the pregnancy experience scale (PES), and version A of the Wijma delivery expectancy/experience questionnaire (WDEQ-A) were completed. In the 36th and 38th weeks of pregnancy, the PES and WDEQ-A questionnaires, as well as birth preference form were further completed. Mode of birth was recorded in the first few days of postpartum. The Fisher's exact test, along with ANOVA and Chi-square tests were used to determine associations between variables. A paired t-test was used to examine within-group comparisons. The Kruskal–Wallis non-parametric test was used to investigate the intervening effect of economic status.

are not available publicly due to the confidentiality of information, but they can be available through the all of the authors (l.amirifarahani@gmail.com, rrr.mousavi@gmail.com, shima_haghani@yahoo.com, ac4733@coventry.ac.uk), ethics committee of Iran University of Medical Sciences (PR@iums.ac.ir), research deputy of Iran University of Medical Sciences (research-m@iums.ac.ir) on reasonable request. Trial Registration: IRCT.ir: IRCT20180427039436N2; https://www.irct.ir/trial/30890.

**Funding:** The present article is extracted from a master thesis funded by the Research Deputy of Iran University of Medical Sciences, Tehran, Iran.

**Competing interests:** The authors have declared that no competing interests exist.

## Results

Post intervention, the mean score of pregnancy experience and FOC did not differ significantly between the three groups. Also, 86.8% of participants in group A, 90.4% of participants in group B, and 62% of participants in the control group preferred to give birth vaginally, which was statistically significant (p = 0.001). Moreover, 66% of participants in group A, 61.5% of participants in group B, and 50% of participants in the control group ultimately gave birth vaginally. None of the participants in group A underwent an elective cesarean section, while this rate was 7.7% and 24% for groups B and control, respectively (p = 0.002).

## Conclusion

Despite the non-significant differences identified between the three groups in terms of pregnancy experience and FOC, prenatal education delivered via social media may be usefully offered to Iranian women keen to receive education flexibly online.

## Trial registration

**Name of the Registry:** Iranian Registry of Clinical Trials. **Trial registration number:** IRCT20180427039436N2. **Date of registration:** 15/06/2018. **URL of trial registry record:** https://www.irct.ir/trial/30890.

## Introduction

The delivery of high quality care during pregnancy is a key goal set by the World Health Organization (WHO) [1]. Childbirth preparation courses contribute toward meeting this goal by improving the lifestyles of those who participate during pregnancy, childbirth, and the postpartum period, as well as protecting the rights of pregnant women through education [2]. For example, by achieving a positive mindset in relation to pain during childbirth through childbirth preparation courses, participants can experience reduced anxiety and are therefore better able to engage in their childbearing journeys [3]. Childbearing preparation courses can also enhance decision making by correcting misconceptions, boosting self-confidence, informing participants in relation to birth choices, reducing the need for pain relief during childbirth, and reducing the fear of childbirth (FOC) [4]. Conversely, poor care delivered during this period can lead to poorer outcomes such as increased rates of cesarean section, postpartum depression, and poor acceptance of maternal role [5].

Aside from education, the extent to which one can adapt to the changes associated with pregnancy can in turn affect ones FOC, and consequently ones birth choices [1]. Pregnant women typically experience some degree of anxiety, fear, and concern in relation to birth, particularly if they have no prior knowledge or experiences of birth [6, 7]. Such FOC is a situational fear which ranges from mild to severe [8]. The prevalence of FOC among Iranian primigravid women is reported to be 80.8% [9]. The reasons for FOC include fear of pain, fear of endangering the health of newborn, misinformation, misconceptions, and negative experiences reported by others [10]. Significantly, FOC (rather than medical need) is one of the most common reasons for cesarean section in Iran [11]. This is significant because cesarean section involves major surgery, and whilst lifesaving in emergency situations, is associated with a myriad of increase risks.

Whilst childbirth preparation courses can significantly reduce FOC and increase self-efficacy [12], studies on their effectiveness have yielded mixed results. For example, in one study, such courses did not have a significant effect on reducing anxiety and increasing self-efficacy [13]. In addition, such courses have been found effective only in reducing labor anxiety and yet ineffective in influencing birth choices [14]. Equally, findings on the effect of perinatal education delivered online using multimedia and virtual methods on FOC are contradictory, as some studies have found them effective in reducing FOC [15], and some have not [16, 17]. Nevertheless, the number of pregnant women seeking pregnancy and childbirth information through online and virtual methods is increasing [18, 19]. Therefore, exploring the efficacy of childbirth preparation courses delivered online may be useful in this context, particularly as the number of cesarean sections has been estimated at 48%-50% in Iran, many of which are performed in response to FOC [11].

Telegram is one of the most popular social media platforms in Iran, enabling voice calling, text chat, group creation, information gathering and entertainment [20]. Despite the growing popularity of social media use among pregnant women, few studies have been conducted on the virtualization of pregnancy and childbirth education. Nevertheless, Tsai et al. (2018) demonstrated that web-based education significantly reduced the stress and increased self-efficacy of pregnant women. Thus, the quality of prenatal care may be improved by integrating common prenatal education methods with internet-based methods [21]. Furthermore, pregnant women, especially nulliparous women are more likely to obtain information related to pregnancy and childbirth from virtual networks and internet resources highlighting the imperative to ensure such information remains accurate and evidence-based [22]. This may be due to the increase in seeking information online more generally. Ghaffari et al. (2017) also reported that, from the mothers' point of view, telegram-based education is more useful than attending in-person breastfeeding classes [20]. Thus, Telegram may be a useful platform upon which to deliver childbirth preparation courses to pregnant women in Iran.

In Iran, attending childbirth preparation courses is not mandatory. Yet such courses have the potential to reduce FOC, and thus potentially birth preferences, experiences and mode of birth, particularly if they are delivered online where an increasing number of pregnant Iranian women are seeking information in this context. Thus, the present study aimed to compare the impact of childbirth preparation courses delivered both in-person and via the Telegram social media platform on pregnancy experience, FOC birth preference, and mode of birth among pregnant women in Iran.

## Methods

### Trial design and participants

This was a quasi-experimental study including two intervention groups receiving social media-based education and in-person education, alongside one control group. This study included primiparous women who had been referred to the prenatal clinic of Milad Hospital in Tehran to receive prenatal care from August to March 2018. Women were invited to participate if they were of Iranian nationality, between 18–35 years old, between 18–20 weeks of pregnancy, had the ability to read and write, and had a device with internet access and the Telegram app downloaded. Women were excluded from participation if they had a history of infertility or were experiencing a high-risk pregnancy or mental illness.

Convenience sampling was used in the recruitment of participants. Participants were blinded from alternate intervention groups and divided into three groups without randomization for practical reasons. If random allocation methods had been used, there may have been contamination between samples, thus decreasing the quality of the study. Group A received

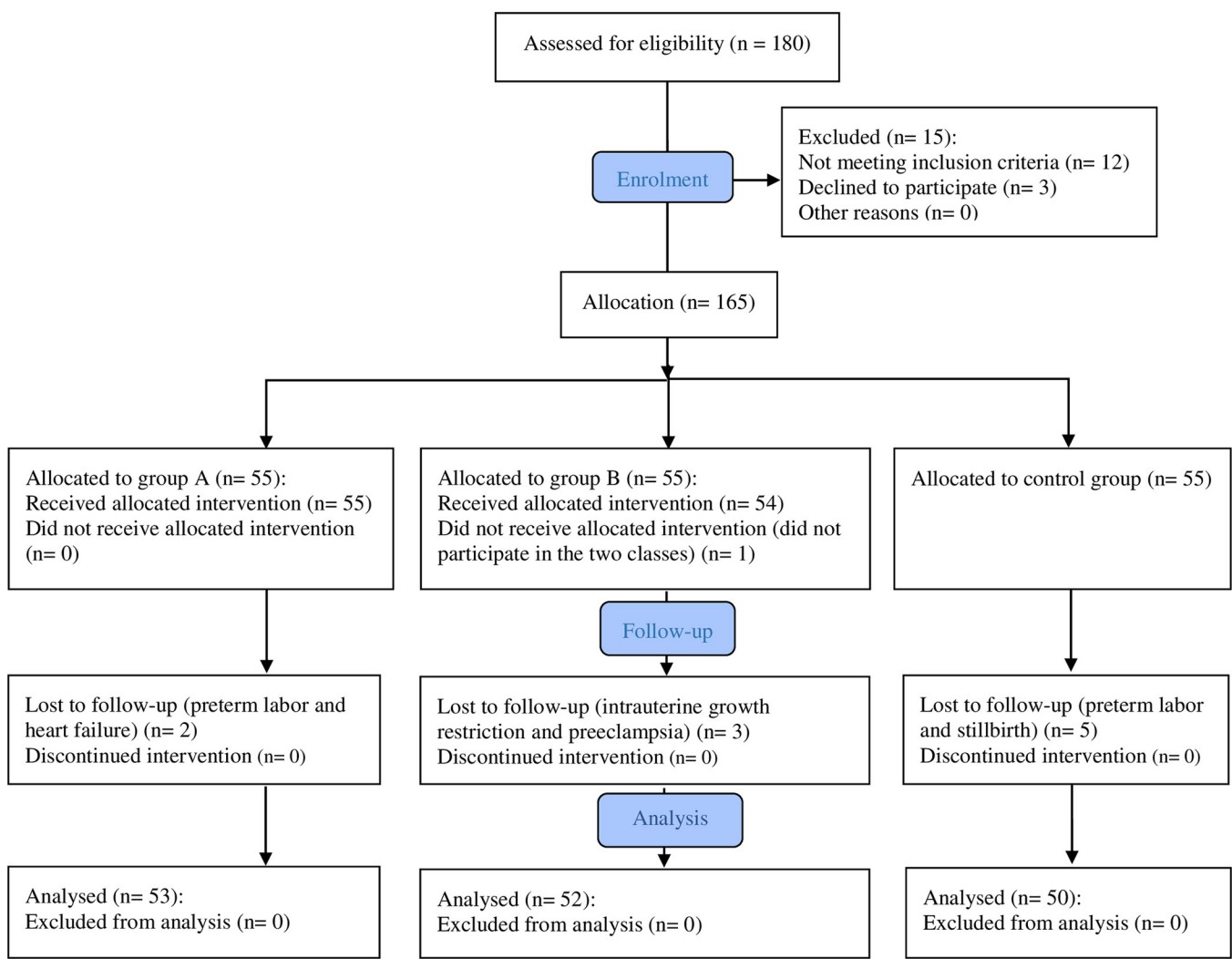

**Fig 1. Flowchart of the recruitment process for participants.**

social media-based education delivered via the Telegram app, and group B received standard in-person education delivered on a weekly basis. Our control group did not receive any childbirth educational intervention. After sampling, participants with preterm labor, any symptoms of high-risk pregnancy, or who chose to withdraw were subsequently excluded. Equally, those in group A who did not engage with the Telegram platform for more than one week, and those in group B who did not attend two classes or more were excluded from the study (withdrawal criteria). Fig 1 outlines our recruitment process.

## Intervention

The content of the both the virtual and in-person educational interventions were delivered in accordance with the national guidelines of the Ministry of Health and Medical Education [3].

For group A, a group called "Virtual Childbirth Preparation Courses" was created within the Telegram app to upload the content of virtual education, answer questions, and exchange ideas. Before joining the group, participants were asked to set up the Telegram app and be online at least once a day to read the messages and provide feedback. The content of virtual

**Table 1. Educational content (groups A and B).**

| Time | Content | Objectives | Uploading the virtual content to the Telegram channel |
|---|---|---|---|
| First session: Between 20–23 weeks of pregnancy | • Personal hygiene with an emphasis on anatomy and physiology during pregnancy | • An introduction to the reproductive system<br>• Changes and adaptations of the body during pregnancy, common complaints, and coping strategies<br>• Personal hygiene | 4 video-casts for theoretical content 9 video clips (1) for exercises during pregnancy 1 podcast for relaxation (1) |
| Second session: Between 24–27 weeks of pregnancy | • Pregnancy diet | • Pregnancy diet with emphasis on what to eat<br>• An introduction to the food pyramid | 1 video-cast for theoretical content 9 video clips (2) for exercises during pregnancy, Repeating relaxation (1) |
| Third session: Between 28–29 weeks of pregnancy | • Mental health during pregnancy | • An introduction to the fetal growth and development<br>• Preparing for motherhood<br>• Preparing for fatherhood | 2 video-casts and 1 PDF file for theoretical content 9 video clips (3) for exercises during pregnancy Repeating relaxation (1) |
| Fourth session: Between 30–31 weeks of pregnancy | • Risk factors during pregnancy | • Learning the signs of preterm birth and how to react to them | 1 video-cast for theoretical content 9 video clips (4) for exercises during pregnancy 1 podcast for relaxation (2) |
| Fifth session: Between 32–33 weeks of pregnancy | • Planning for birth and selecting the type/mode of birth | • Physiological birth vs. cesarean section<br>• Different pain control methods during labor<br>• Selecting the location for birth and necessary equipment | 2 video-casts and 1 PDF file for theoretical content Repeating relaxation (2) |
| Sixth session: Between 34–35 weeks of pregnancy | • An introduction to physiological vaginal birth | • An introduction to birth hormones<br>• An introduction to labor stages and self-care at each stage | 1 video-cast, 1 PDF file, and 2 video clips for theoretical content 11 video clips for labor and childbirth 1 podcast for breathing techniques 1 podcast for relaxation techniques (3) |
| Seventh session: At 36 weeks of pregnancy | • Postpartum care and breastfeeding | • Postpartum care and recognition of dangerous symptoms<br>• An introduction to breastfeeding methods and breast diseases<br>• An introduction to postpartum exercises | 1 video-cast, 1 PDF file, and 8 pictures for theoretical content 8 video clips for massage during pregnancy and labor Repeating relaxation (3) |
| Eighth session: At 37 weeks of pregnancy | • Neonatal care | • Neonatal care and risk factors | 2 PDF files and 2 videos for theoretical content 9 video clips for postpartum exercises Repeating relaxation (1) |

education was designed based on Meyer's multimedia principles [23], in the form of text, image, podcast, video-cast and video clip in MPEG-4 (MP4) format. All theoretical and practical education delivered were designed as distance learning files. The maximum size of these files was 50 MB delivered in the form of 5–15-minute videos. To prevent the sudden upload of content and increase the quality of education, the content of each session was uploaded in the divided sections daily (except Thursday and Friday) during the time allocated for that session. Videos were then made available to watch on repeat. To ensure that the correct breathing and relaxation techniques were used, two 2-hour in-person sessions were organized in addition to the online educational intervention.

The childbirth preparation courses delivered to group B were held in-person at the hospital according to the national guidelines of the Ministry of Health and Medical Education. Courses were delivered via eight 2-hour sessions. During each session, 1 hour was devoted to theoretical topics; 45 minutes were devoted to stretching exercises, breathing and relaxation techniques, and practical education on correcting position and massage, whilst 15 minutes were devoted to questions and answers. The educational content, time, and objectives of the courses in both groups A and B were the same and are displayed in Table 1.

Participants in the control group did not attend any of the childbirth preparation courses held in or out of the hospital. Our control group only received routine prenatal care as did groups A and B in equal number. All participants were followed and supported until they had given birth.

## Instruments/Outcomes

The demographic information questionnaire measured the variables of age, age at the time of marriage, the level of education and employment status of couples, economic status, body mass index, insurance status, and recent pregnancy status.

The Brief version of the Pregnancy Experience Scale (PES), first developed by DiPietro (2008), was prepared with 20 items used to measure pregnancy experience. The first 10 items measured the 'uplifts', and the second 10 items indicated the 'hassles' associated with pregnancy. Participants were invited to complete a 4-option Likert scale, with answer options ranging from not at all, somewhat, quite a bit, and a great deal, with the score of zero to 3 respectively. There was a minimum score of zero and a maximum score of 30 for each subscale. To calculate the pregnancy experience, the total score of 'hassles' was first divided by the total score of 'uplifts'. If the result was less than one, this indicated more a uplifting feeling than a feeling of hassle and vice versa (if the result was more than one, this indicated more hassle than uplift).

Cronbach's alpha of the English version of the scale was 0.82 for the uplift's subscale and 0.83 for the hassle's subscale. Time stability was measured at 0.56–0.83 by the test-retest method [24]. Cronbach's alpha of the Persian version of this scale was 0.77 for the uplift's subscale and 0.67 for the hassle's subscale. Additionally, the intra-class correlation coefficient (ICC) was 0.711 for the uplift's subscale and 0.67 for the hassle's subscale [25]. The reliability of this questionnaire has been calculated with a Cronbach's alpha of 0.66 for the uplift's subscale and 0.7 for the hassle's subscale.

The Wijma Delivery Expectancy / Experience Questionnaire (WDEQ-A) was used to measure FOC. The WDEQ-A version has 33 items based on a 6-option Likert questionnaire ranging from ' not at all. . .' to 'extremely. . .' with total score of 0 to 165. A score of 37 or less is indicative of "mild fear", a score of 38–65 refers to "moderate fear", a score of 66–84 indicates "severe fear" and a score of 85 or higher is indicative of "clinical fear". The reliability of the questionnaire has also been confirmed with a Cronbach's alpha of 0.89. The reliability of the two halves of the test is 0.91 [26]. The reliability of the Persian version of this questionnaire in Iran has been reported with a Cronbach's alpha of 0.64 [27]. The reliability of this questionnaire has been calculated with a Cronbach's alpha of 0.85.

We also sought to explore outcomes in relation to birth preferences and mode of birth. As such, during the 18[th] and 20[th] weeks of pregnancy, demographic information questionnaire, the PES, and WDEQ-A questionnaires were completed. Yet in the 36[th] and 38[th] weeks of pregnancy, as well as the PES and WDEQ-A questionnaires being completed, participants were also asked to self-report their birth preferences. Mode of birth was also recorded in the first few days following childbirth.

## Sample size

To determine the required sample size at the significance level of 0.05 and the test power of 80%, we assumed that the effect of education within each of the two interventions compared to the control group would reduce the rate of cesarean sections in pregnant women by 25%. The

following formula was used to calculate our sample size;

$$n = \frac{(z_1 - {}^{\alpha}/_2 \sqrt{2\bar{p}\bar{q}} + z_{1-\beta}\sqrt{p_1 q_1 + p_2 q_2})^2}{(p_1 - p_2)^2} = \frac{[(1.96 \times \sqrt{2 \times 0.42 \times 0.57}) + (0.84 \times \sqrt{(0.55 \times 0.45) + (0.3 \times 0.7)})]^2}{(0.55 - 0.3)^2} = 50$$

$$\bar{p} = \frac{p_1 + p_2}{2}$$

In the present study, the ratio of primiparous women who give birth by cesarean section in Iran was assumed to be 0.55 based on available statistics [28]. The sample size in each group was also estimated to be 55 participants considering a 10% sample drop. This is more than the sample size calculated by Toohill et al. [29], where at 95% confidence level, 80% test power, the accuracy of 13 and standard deviation of 21.9 the sample size was reportedly set at 45 participants, based on mode of birth.

## Statistical analysis

Data analysis was performed using SPSS software version 19, via descriptive and inferential statistics. Descriptive statistics such as numerical indicators and frequency distribution tables were also used to make sense of the data. The Fisher's exact test, along with ANOVA and Chi-square tests were used to determine associations between variables in the three groups. A paired t-test was used to examine within-group comparisons. Lastly, the Kruskal–Wallis non-parametric test was used to investigate the intervening effect of economic status in the areas of uplifts and hassles in relation to participants' experience of pregnancy. In all tests, the significance level was less than 0.05.

## Ethical considerations

Ethical approval was granted for the present study by the Research Deputy of Iran University of Medical Sciences (Project code: IR.IUMS.REC1396.9511373011). This study has also been registered in the Iranian clinical trial registry via the following code: IRCT201804447070394436N2. Written informed consent was obtained from all study participants. Respondents were also fully informed of the study purpose and procedures. They were assured confidentiality throughout, and that they could leave the study at any time without giving reason.

## Results

A total of 180 participants were evaluated for their eligibility to participate in the study, from whom 165 were eligible and included. Overall, 55 were allocated to group A, 55 were allocated to group B, and 55 were allocated to the control group. The overall number of participants excluded from the study during follow-up and the final number of participants included in our statistical analysis are presented in Fig 1.

The characteristics of participants alongside the results of our comparative statistical analyses are presented in Table 2. Apart from the two variables relating to the employment status of spouse and economic status, there were no significant differences identified in terms of individual characteristics between the three groups. Statistical tests demonstrated that the variables of the employment status of spouse and economic status did not have a significant relationship with the experience of pregnancy (in two subscales of uplifts and hassles) and FOC. They were therefore considered to be non-intervening.

According to the results of our one-way ANOVA test, there was no significant between-group difference identified in terms of pregnancy experience and FOC. However, results of the

**Table 2. Individual characteristics of study participants and comparisons between social media-based education through Telegram app, in-person education in childbirth preparation courses, and control groups (n = 155).**

| Characteristic/Group | Social media-based delivery (n = 53) | In-person delivery (n = 52) | Control (n = 50) | P value* |
|---|---|---|---|---|
| Age (year), mean (SD) | 25.67 ± 4.79 | 27.59 ± 3.61 | 26.52 ± 4.35 | 0.07 |
| Age of marriage (year), mean (SD) | 22.64 ± 4.77 | 24.3 ± 3.87 | 23.66 ± 4.5 | 0.15 |
| Level of education, n (%) | | | | 0.09 |
| Elementary and middle school | 1 (1.9) | 0 | 1 (2) | |
| High school | 3 (5.7) | 0 | 4 (8) | |
| Diploma | 23 (43.4) | 15 (28.8) | 19 (38) | |
| Collegiate | 26 (49.1) | 37 (71.2) | 26 (52) | |
| Level of education of couple, n (%) | | | | 0.29 |
| Elementary and middle school | 3 (5.7) | 0 | 3 (6) | |
| High school | 4 (7.5) | 3 (5.8) | 2 (4) | |
| Diploma | 28 (52.8) | 21 (40.4) | 24 (48) | |
| Collegiate | 18 (34) | 28 (53.8) | 21 (42) | |
| Employment status, n (%) | | | | 0.64 |
| Housewife | 46 (86.8) | 40 (76.9) | 40 (80) | |
| Employee | 6 (11.3) | 9 (17.3) | 9 (18) | |
| Self-employment | 1 (1.9) | 3 (5.8) | 1 (2) | |
| Employment status of couple, n (%) | | | | **0.02**[**] |
| Unemployed | 1 (1.9) | 1 (1.9) | 0 | |
| Worker | 13 (24.5) | 3 (5.8) | 14 (28) | |
| Employee | 16 (30.2) | 16 (30.8) | 9 (18) | |
| Self-employment | 23 (43.4) | 32 (61.5) | 27 (54) | |
| Economic status, n (%) | | | | **0.03**[***] |
| Undesirable | 5 (9.4) | 3 (5.8) | 3 (6) | |
| Relatively desirable | 37 (69.8) | 35 (67.3) | 22 (44) | |
| Desirable | 11 (20.8) | 14 (26.9) | 24 (48) | |
| Rich | 0 | 0 | 1 (2) | |
| Recent pregnancy status, n (%) | | | | 0.78 |
| Wanted | 43 (81.1) | 40 (76.9) | 41 (82) | |
| Unwanted | 10 (18.9) | 12 (23.1) | 9 (18) | |
| Body mass index (kg/m$^2$),mean (SD) | 24.97 ± 4.26 | 24.78 ± 3.96 | 23.33 ± 3.39 | 0.07 |
| Insurance status, n (%) | | | | 0.1 |
| Yes | 50 (94.3) | 52 (100) | 50 (100) | |
| No | 3 (5.7) | 0 | 0 | |

* p < 0.05 is significant.

**** According to the ANOVA test, the employment status of spouse did not have significant relationship with pregnancy experience in two subscales of uplifts (p = 0.95) and hassles (p = 0.5) and FOC (p = 0.49) and based on Fisher's exact test, the employment status of couple with birth preference (P = 0.23) and the mode of birth (p = 0.96), so it was considered non-intervening.

*** According to the Kruskal–Wallis test, the economic status did not have significant relationship with pregnancy experience in two subscales of uplifts (p = 0.15) and hassles (p = 0.79) and FOC (p = 0.16) and based on Fisher's exact test, the economic status with birth preference (P = 0.46) and the mode of birth (p = 0.06), so it was considered non-intervening.

within-group paired t-test demonstrated that the experienced uplifts in pregnancy in group A along with the control group was significantly increased, whereas the mean score of FOC in group B was significantly reduced post intervention (Table 3).

Due to the significant results of the Chi-square test in relation to the outcomes of birth preference and modes of birth, groups were evaluated in pairs. Results demonstrated that

**Table 3. Within-group and between-group comparisons of pregnancy experience and FOC in three groups before and after the intervention.**

| Group | | Social media-based delivery (n = 53) | In-person delivery (n = 52) | Control (n = 50) | P value ANOVA test* |
|---|---|---|---|---|---|
| **Pregnancy experience** (Mean ± SD) | Before | 0.76 ± 0.26 | 0.69 ± 0.2 | 0.72 ± 0.25 | 0.3 |
| | After | 0.71 ± 0.24 | 0.7 ± 0.21 | 0.72 ± 0.19 | 0.96 |
| | P value Paired t-test** | 0.13 | 0.72 | 0.82 | |
| Uplifts (Mean ± SD) | Before | 23.92 ± 3.61 | 24.9 ± 3.56 | 24.18 ± 3.33 | 0.34 |
| | After | 24.92 ± 2.83 | 25.65 ± 2.98 | 25.54 ± 2.82 | 0.97 |
| | P value Paired t-test | **0.02** | 0.1 | **0.002** | |
| Hassles (Mean ± SD) | Before | 17.83 ± 4.65 | 17.05 ± 5.18 | 17.06 ± 4.5 | 0.63 |
| | After | 17.41 ± 5.18 | 18.03 ± 5.49 | 18.16 ± 4.4 | 0.72 |
| | P value Paired t-test | 0.56 | 0.26 | 0.15 | |
| **Fear of childbirth** (Mean ± SD) | Before | 53.77 ± 24.62 | 53.38 ± 16.01 | 50.16 ± 18.44 | 0.61 |
| | After | 50.9 ± 23.75 | 47.96 ± 16.14 | 53 ± 20.08 | 0.45 |
| | P value Paired t-test | 0.19 | **0.01** | 0.21 | |

* Between- group comparison

** Within- group comparison

participants in groups A and B had a higher preference for giving birth vaginally (86.8% and 90.4%, respectively) than the control group (62%). However, in relation to mode of birth, post-partum follow-up indicated a significant effect of group A on group B and the control group. Yet as presented in Table 4, no significant difference in relation to mode of birth was identified in group B when compared to the control group.

## Discussion

The present study compared the effect of delivering childbirth preparation courses via two different modalities (in person and via social media) on pregnancy experience, FOC, birth preference, and mode of birth in pregnant women. Findings identify that neither method improved the pregnancy experience when compared to the control group. Yet social media-based education delivered via the Telegram app (group A) was able to significantly increase 'uplifts' in pregnant women and thus improve the experience of pregnancy, although in a non-significant way.

Similarly, Wu and Hung (2019) examined the effect of prenatal education through Facebook on pregnant women's well-being and identified no significant effect from the virtual

**Table 4. Between-group and within-group comparison of birth preference and mode of birth in three groups.**

| Group | | Social media-based delivery (n = 53) | In-person delivery (n = 52) | Control (n = 50) | P value (Between groups) | *P value (A-B groups) | *P value (B-C groups) | *P value (A-C groups) |
|---|---|---|---|---|---|---|---|---|
| **Birth preference, n (%)** | Vaginal delivery | 46 (86.8) | 47 (90.4) | 31 (62) | **0.001** | **< 0.001** | **0.001** | **0.003** |
| | Cesarean section | 7 (13.2) | 5 (9.6) | 19 (38) | | | | |
| **Mode of birth, n (%)** | Vaginal delivery | 35 (66) | 32 (61.5) | 25 (50) | **0.002** | **< 0.001** | **0.001** | 0.07 |
| | Cesarean section | 18 (34) | 16 (30.8) | 13 (26) | | | | |
| | Elective cesarean section | 0 | 4 (7.7) | 12 (24) | | | | |

*To compare pairwise groups and perform multiple tests; Bonferroni correction was used. P value less than 0.167 was considered significant.

intervention on pregnant women's well-being [21]. Yet a different study exploring the effectiveness of a web-based prenatal education program when compared to routine care demonstrated that such a program was able to significantly reduce pregnancy stress. The reason for this inconsistency may be attributed to the synergistic effect of delivering both prenatal education and prenatal care together in a virtual manner [20].

In the present study, between-group results revealed no significant difference in the mean score of FOC. However, the mean score of FOC decreased in the groups A and B with increasing gestational age, and yet this decreased in the control group. Hence, group A was able to prevent the increase in FOC closer to the time of birth and reduce the severity of FOC. Nevertheless, these changes remained statistically insignificant. Likewise, Bergstrom et al. (2010) were unable to demonstrate the superiority of prenatal education delivered via combined multimedia and in-person education in comparison with only in-person education with a booklet in reducing FOC [16]. Similarly, the findings of Nair et al. (2015) could not demonstrate that childbirth preparation courses delivered via video to primiparous women reduced FOC [17]. In contrast, the study of Isbir et al. (2016) demonstrated the superiority of combined multimedia and in-person education on the FOC when compared to routine prenatal care [15]. Nevertheless, prenatal education can fail to focus on the psychological dimensions of pregnancy and childbirth, a key factor in reducing FOC [30]. As such, the study by Isbir et al (2016) may have been able to evidence reductions in FOC by providing education in both physical and psychological dimensions. Such combinations of routine and multimedia methods may therefore be a key factor in achieving more favorable results in future.

Indeed, according to the systematic review and meta-analysis of Hosseini et al. (2018), both educational methods in the physical dimension in the form of childbirth preparation courses and also in psychological dimension in the form of group psycho-education and telephone psycho-education counseling can be effective in reducing FOC [31, 32]. As such, it will be important to consider both dimensions in the development of future prenatal educational courses. Yet it will also be important to consider that if the educational content of such courses is designed inadequately, FOC may be increased, especially in vulnerable women [33–35].

In the present study, groups A and B were shown to have the highest preference for giving birth vaginally. Study groups A and B also had the highest number of vaginal births, whereas the control group had the highest number of elective cesarean sections. Meanwhile, none of the participants in group A had an elective cesarean section. In addition, a paired comparison of the groups revealed that participants in groups A and B had higher preference for giving birth vaginally compared to the control group. Furthermore, group B had a higher preference for giving birth vaginally compared to group A. However, group A had a significantly higher vaginal birth rate than group B and the control group. Whilst such findings are encouraging, larger randomized trials of prenatal educational interventions which include combined phycological and physical dimensions included may yield more significant results.

Results presented by Kulkarni et al. (2014) on the effect of a web-based education in relation to the advantages and disadvantages of vaginal birth and the cesarean section on the birth preferences confirms the results of the present study [36]. In addition, the results of another study confirmed that a quarter of pregnant women who did not receive prenatal education preferred to give birth via cesarean section and another 30% were seeking information on cesarean section, believing cesarean section to be safer than giving birth vaginally [37]. Encouragingly, the results of the present study demonstrate that by increasing awareness in relation to giving birth vaginally and via cesarean section women's preference for cesarean section can be reduced. Elsewhere, multimedia prenatal education has also been evidenced to increase knowledge and thus positively contribute in similar ways [38].

Ultimately, prenatal education can empower pregnant women to make informed decisions by providing evidence-based information [32]. Nevertheless, participation in childbirth preparation courses in Iran is not mandatory, and many Iranian pregnant women obtain information about pregnancy and childbirth online using social media, which in many cases may not be accurate. Social media, may therefore be an important tool that maternity educators can use to disseminate evidence-based educational materials to pregnant women [39]. It is also possible that a mixture of online and face to face classes may be more manageable for pregnant women who have other children or who must travel. In this regard, it may be most useful to develop educational content in accordance with the conditions and needs of society. Such information could usefully be easy to access and developed to satisfy the needs of pregnant women through evidence-based virtual learning [40].

## Limitations

Non-randomized studies such as ours are more prone to systematic and confounding biases than randomized clinical trials; consequently, it has been difficult to make causal inferences about the effect of our intervention [41]. The lack of participation of half of the participants in the study group A in the in-person visiting of the delivery room was the other of the limitations of this study. To compensate for this, an attempt was made to prepare a video clip from the delivery room and the women watched it. Future research could usefully examine the satisfaction of pregnant women engaged in prenatal education delivered via social media, compare the impact of integrated social media-based and in-person education with only in-person education, and compare the impact of psychological education with social media-based and in-person education on the experience of pregnancy, FOC and mode of birth. Other limitations of the study included participants' potential to obtain pregnancy and childbirth information from other sources during pregnancy, conceivably affecting the accuracy of results.

## Conclusion

The present study identified that the participants who received prenatal education delivered via social media experienced uplifts during pregnancy, did not prefer to give birth via elective cesarean section and eventually gave birth vaginally. In addition, although there were no significant results regarding reductions in FOC, the present study demonstrated that women who did not receive any prenatal education became more afraid of labor and birth with increasing gestational age. Yet prenatal education delivered via social media was able to either address the fears of pregnant women or reduce its severity. As such, prenatal education could usefully be designed in accordance with the conditions and needs of society with additional options for delivery via social media.

## Supporting information

**S1 Checklist.**
(DOC)

**S1 Protocol.**
(PDF)

**S2 Protocol.**
(PDF)

## Acknowledgments

The author would like to thank the Milad Hospital's staff and the pregnant women who participated in this study.

## Author Contributions

**Conceptualization:** Leila Amiri-Farahani.

**Data curation:** Leila Amiri-Farahani.

**Formal analysis:** Shima Haghani.

**Funding acquisition:** Leila Amiri-Farahani.

**Methodology:** Seyedeh Robab Mousavi, Shima Haghani.

**Software:** Seyedeh Robab Mousavi.

**Supervision:** Leila Amiri-Farahani.

**Validation:** Leila Amiri-Farahani.

**Writing – original draft:** Seyedeh Robab Mousavi.

**Writing – review & editing:** Sally Pezaro.

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
