## [Decision Letter · Decision Letter 0]

29 Mar 2022

PONE-D-21-05419

Comparing the effect of childbirth preparation courses by two methods of in-person training and social media-based training on pregnancy experience, fear of childbirth, and type of delivery in pregnant women: A quasi-experimental study

PLOS ONE

Dear Dr. Amiri-Farahani,

Thank you for submitting your manuscript to PLOS ONE. After careful consideration, we feel that it has merit but does not fully meet PLOS ONE’s publication criteria as it currently stands. Therefore, we invite you to submit a revised version of the manuscript that addresses the points raised during the review process.

The reviewers raised a number of issues, including their concern about the appropriateness and clarity in your choice of study design. They raised concerns about the methodological/statistical approach, as well as with the English grammar and language usage in the manuscript. Their concerns can be viewed in full, below and in the attached file.

We look forward to receiving your revised manuscript.

Kind regards,

Natasha McDonald, PhD

Associate Editor

PLOS ONE

Journal Requirements:

3. Thank you for including your ethics statement: "The present study, while obtaining the code of ethics with the number: IR.IUMS.REC1396.9511373011 from the Research Deputy of XX University of Medical Sciences has been registered in the Iranian clinical trial registry with the code: IRCT201804447070394436N2. A written informed consent was obtained from all the study participants."

4. Thank you for stating the following in your Competing Interests section: "there is no competing of interest"

Reviewers' comments:

Reviewer's Responses to Questions

**Comments to the Author**

1. Is the manuscript technically sound, and do the data support the conclusions?

Reviewer #1: No

Reviewer #2: Yes

Reviewer #3: Yes

2. Has the statistical analysis been performed appropriately and rigorously? 

Reviewer #1: No

Reviewer #2: I Don't Know

Reviewer #3: Yes

3. Have the authors made all data underlying the findings in their manuscript fully available?

Reviewer #1: No

Reviewer #2: No

Reviewer #3: Yes

4. Is the manuscript presented in an intelligible fashion and written in standard English?

Reviewer #1: No

Reviewer #2: No

Reviewer #3: No

5. Review Comments to the Author

Reviewer #1: The abstract is entirely unclear: Example:

Pregnant women are more interested to obtain information from internet sources, so this study aimed to compare the effect of childbirth preparation courses by two methods of in-person and social media-based education on pregnancy experience, fear of childbirth, birth preference, and type of delivery.

Unclear content!

Another example:

Sampling was done by convenience method and the samples were divided into three groups.

Unclear, grammar errors.

The entire abstract needs to be rewritten because it is unclear and contains many grammar errors.

The paper needs to be carefully checked by a native speaker because aside from content related problems also the presentation in English is very difficult to read.

parallel quasi-experimental study: How is this defined? Add a citation if this is a standard term, otherwiese make it clear that you created this term and explain it.

Table 2: Individualcharacteristicsofstudyparticipantssandcomparisons

What is 's'?

The authors apply multiple tests, however, no multiple testing correction is applied. I suggest Bonferroni (cite the ref)

https://www.mdpi.com/2504-4990/1/2/39

Reviewer #2: Abstract: Drop " PES and WDEQ-A" and replace with " Pregnancy Experience Scale" and Wijma Delivery Expectancy / Experience Questionnaire since the acronym isn't used later in the abstract.

Abstract: -The authors should indicate to which of the interventions A and B were attributed.

-page 6: A sentence or two should be included to provide possible justifications for the mixed results of the previous studies. Will different approaches work better in different settings?

-Page 7: It would be helpful to explain why nulliparous women are more likely to obtain information from virtual networks.

-Page 8: Provide a detailed description of how the women were allocated into 3 groups.

- The authors mentioned that the objective of this research was to compare the effect of childbirth preparation courses by two methods of in-person and social media-based education, however, it seems that participants of group A received a mixed intervention; they were provided social media-based training and a two 2-hour sessions in-person training in the hospital. Please clarify.

- It is important that the authors check each reference carefully against the original publication to ensure the information is accurate and relevant. there are several cases that the references that have not been used carefully for example on page 5, in reference 10 it is no data regarding the prevalence of FOC, and reference 5 did not provide adverse outcomes of poor care of pregnant women.

- More information is needed on how the PES and WDEQ-A questionnaires were used in this study; how the authors assessed the reliability of these tools?

- The methodological limitations of the study need to be clearly stated.

- I would suggest that the manuscript undergo a careful review for English grammar and sentence structure. Also a more appropriate use of terminology is needed.

Reviewer #3: PlosONE REVIEW Carmen Power 22.2.22

Short title: Social media-based training and pregnancy and childbirth outcome

ABSTRACT – note the whole text here is repeated twice

Results section: Give percentages of vaginal birth and elective CS for different study groups otherwise the conclusion doesn’t make sense as it’s not a summary of the results section.

*Ethical permission and written consent – yes

SUMMARY

Expand limitations section – Was the drop-out rate different between groups? If so, this in itself could be a suggestion that women prefer to attend in-person/online pregnancy/birth prep classes.

INTRODUCTION

Check meaning in sentence 2 – repeat of childbirth preparation courses after ‘provide’.

METHOD

Good use of validated questionnaires. Description of statistical methods is good.

A few things to think about and possibly discuss:

1. I’m not sure I understand why the women were ‘non-randomly’ allocated to the 3 groups? How did this prevent their awareness of the other groups if they were choosing their group? And think about how might this have affected the results as it could mean that women who wanted a vaginal birth were more likely to choose a certain group and vice versa. Having studied the flow chart, perhaps you meant to say they were randomly allocated to groups? (i.e. neither the participant nor researcher could show any bias)

2. Could the content involved in online prep (5-15 mins) sessions be linked to it being available on a daily basis so women could watch videos repeatedly, remembering and reinforcing the information?

3. Did Group A having a 2 x 2 hour face to face component blur the findings at all?

4. Did Group B also receiving videos and music blur the difference between the 2 groups?

RESULTS

1. Think about showing your full statistical workings and outcomes in the tables.

2. Clarify in Table 2 (and other tables where needed) whether bracketed numbers are the standard deviation or a percentage of the non-bracketed number directly before.

3. P19 – text underneath table 4 is confusing – please clarify your meaning here – what did the statistical outcomes signify?

DISCUSSION

1. The findings are important in that FOC is a crippling psychological condition that can results in unnecessary interventions. Could a larger sample size be used in a future study?

2. It might also be mentioned that a mixture of online and face to face classes is more manageable for pregnant women who have other children or who have to travel.

GENERAL

Ideally this paper needs to be proofread and edited by a fluent English speaker as there are differences in grammar and turns of phrase that interfere with a smooth reading process. More importantly, the semantics are sometimes lost and your results could be misinterpreted during reading although ultimately they become clear.

6. PLOS authors have the option to publish the peer review history of their article (what does this mean?). If published, this will include your full peer review and any attached files.

Reviewer #1: No

Reviewer #2: No

Reviewer #3: **Yes: **Dr Carmen Power

---

## [Author Response · Author response to Decision Letter 0]

16 May 2022

Dear Editor and reviewers,

Thank you so much for your valuable comments and time that you spend for revision. Your comments and insights have improved the quality of our manuscript extensively. Please see how we have responded to each of the comments below: 

Reviewers' comments:

Reviewer #1: The abstract is entirely unclear: Example:

Pregnant women are more interested to obtain information from internet sources, so this study aimed to compare the effect of childbirth preparation courses by two methods of in-person and social media-based education on pregnancy experience, fear of childbirth, birth preference, and type of delivery.

Unclear content!

Another example:

Sampling was done by convenience method and the samples were divided into three groups.

Unclear, grammar errors. The entire abstract needs to be rewritten because it is unclear and contains many grammar errors. The paper needs to be carefully checked by a native speaker because aside from content related problems also the presentation in English is very difficult to read. Answer: abstract and full text was edited by a native speaker. 

Parallel quasi-experimental study: How is this defined? Add a citation if this is a standard term, otherwise make it clear that you created this term and explain it. Answer: actually, it was not a specific term. It was my mistake in edition. This part was edited (page 8, line 5-6). 

Table 2: Individual characteristics of study participants and comparisons

What is 's'? Answer: thank you so much for your attention. The sentence was edited (page 17).

The authors apply multiple tests; however, no multiple testing correction is applied. I suggest Bonferroni (cite the ref). https://www.mdpi.com/2504-4990/1/2/39.. Answer: we did not use any multiple tests in between group comparisons about pregnancy experience and fear of childbirth because there were not statistical significant between groups in table 3. In table 4, description about Bonferroni correction was added (page 19 line 3). 

Reviewer #2: Abstract: Drop " PES and WDEQ-A" and replace with " Pregnancy Experience Scale" and Wijma Delivery Expectancy / Experience Questionnaire since the acronym isn't used later in the abstract. Answer: it was edited (page 3 line 16-17). 

Abstract: -The authors should indicate to which of the interventions A and B were attributed. Answer: it was edited (page 3 line 13-14). 

-page 6: A sentence or two should be included to provide possible justifications for the mixed results of the previous studies. Will different approaches work better in different settings? Answer: Sentences were added regarding the contradictions of the studies (page 6 line 19-21).

-Page 7: It would be helpful to explain why nulliparous women are more likely to obtain information from virtual networks. 

Answer: All pregnant women, after being informed of their pregnancy, seek information about pregnancy and childbirth through virtual and online networks; this problem is more pronounced in novice women who are experiencing pregnancy and childbirth for the first time. We have added information here (page 7 line 11-14).

-Page 8: Provide a detailed description of how the women were allocated into 3 groups. Answer: As on page 8 line 13-19 in section Trial Design and Participants is described, the individuals were divided into three groups in a non-random method. 

It should be noted that the number of people participating in childbirth preparation classes in Iran is approximately 1 in 5 pregnant women. If sampling was done in centers with low perinatal care, the minimum duration of sampling was estimated to be more than 2 years. This was not considered feasible because this study was conducted as part of a master's degree program. We selected Milad prenatal care clinic in Tehran because it is the largest provider of services and care during pregnancy and has many clients. If we had used random allocation methods, there may have been contamination between samples, thus decreasing the quality of the study. For this reason, according to the mentioned reasons, non-random sampling method was used.

Further reasons to select one perinatal clinic (Milad) and a non-random approach include: 

• The difference between trainers: (each perinatal care clinic has specific trainers and there are definitely different trainers on them and due to individual differences may not have the same training if we select different perinatal care clinic. 

• Socio-economic differences between pregnant women in different geographical areas. 

• The low number of people participating in prenatal classes eligible in other prenatal clinics.

- The authors mentioned that the objective of this research was to compare the effect of childbirth preparation courses by two methods of in-person and social media-based education, however, it seems that participants of group A received a mixed intervention; they were provided social media-based training and a two 2-hour sessions in-person training in the hospital. Please clarify. Answer: Thank you for your valuable comment. As described on page 11-12, individuals in both training groups received similar training content, except that in the social media-based group through the virtual network-based program and in the in-person group through face-to-face training in the classroom. This question is probably due to text errors that were edited.

- It is important that the authors check each reference carefully against the original publication to ensure the information is accurate and relevant. there are several cases that the references that have not been used carefully for example on page 5, in reference 10 it is no data regarding the prevalence of FOC, and reference 5 did not provide adverse outcomes of poor care of pregnant women. Answer: it was edited (pages 5-6).

- More information is needed on how the PES and WDEQ-A questionnaires were used in this study; how the authors assessed the reliability of these tools? Answer: it was edited (pages 14 line 5-7).

- The methodological limitations of the study need to be clearly stated. Answer: based on the other reviewers' comment, this part was edited (page 23 line 4-16). 

- I would suggest that the manuscript undergo a careful review for English grammar and sentence structure. Also a more appropriate use of terminology is needed. Answer: the whole of manuscript was edited by a native speaker. 

Reviewer 3 

Short title: Social media-based training and pregnancy and childbirth outcome

Reviewer reports:

ABSTRACT:

1. note the whole text here is repeated twice. Answer: Based on your comment, this part was edited.

2. Results section: Give percentages of vaginal birth and elective CS for different study groups otherwise the conclusion doesn’t make sense as it’s not a summary of the results section. Answer: Based on your comment, this part was edited (highlighted by yellow color at page 4 line 4-7).

3. Ethical permission and written consent – yes. Answer: The ethical permission and written consent was edited (highlighted by yellow color at page 16 line 3-10).

SUMMARY:

Expand limitations section – Was the drop-out rate different between groups? If so, this in itself could be a suggestion that women prefer to attend in-person/online pregnancy/birth prep classes. Answer: The limitations section were expanded (highlighted by yellow color at page 5 and 23). The drop-out rate was similar between groups. 

INTRODUCTION

Check meaning in sentence 2 – repeat of childbirth preparation courses after ‘provide’.

Answer: Based on your comment, this part was edited (highlighted by yellow color at page 5).

METHOD

Good use of validated questionnaires. Description of statistical methods is good.

A few things to think about and possibly discuss:

I’m not sure I understand why the women were ‘non-randomly’ allocated to the 3 groups? How did this prevent their awareness of the other groups if they were choosing their group? And think about how might this have affected the results as it could mean that women who wanted a vaginal birth were more likely to choose a certain group and vice versa. Having studied the flow chart, perhaps you meant to say they were randomly allocated to groups? (i.e. neither the participant nor researcher could show any bias). Answer: As on page 10 in flow chart is showed, the individuals were divided into three groups and not random allocated. In the present study, we were not able to randomly allocated to groups for the reasons mentioned below: 

It should be noted that the number of people participating in childbirth preparation classes in Iran is approximately 1 in 5 pregnant women. If sampling was done in centers with low perinatal care, the minimum duration of sampling was estimated to be more than 2 years. This was not considered feasible because this study was conducted as part of a master's degree programme. We selected Milad prenatal care clinic in Tehran because it is the largest provider of services and care during pregnancy and has many clients. If we had used random allocation methods, there may have been contamination between samples, thus decreasing the quality of the study. For this reason, according to the mentioned reasons, non-random sampling method was used.

Further reasons to select one perinatal clinic (Milad) and a non-random approach include: 

- The difference between trainers: (each perinatal care clinic has specific trainers and there are definitely different trainers on them and due to individual differences may not have the same training if we select different perinatal care clinic. 

- Socio-economic differences between pregnant women in different geographical areas. 

- The low number of people participating in prenatal classes eligible in other prenatal clinics.

1. Could the content involved in online prep (5-15 mins) sessions be linked to it being available on a daily basis so women could watch videos repeatedly, remembering and reinforcing the information? Answer: Yes, it was possible, women can watch videos repeatedly, remembering and reinforcing the information (highlighted by yellow color at page 11 line 13-16).

2. Did Group A having a 2 x 2 hour face to face component blur the findings at all? Answer: Yes, by holding these meetings, the problems were identified and corrected. It was ensured that there was no difference between the two groups in terms of performing correct stretching and corrective movements (highlighted by yellow color at page 11, lines 14-16).

3. Did Group B also receiving videos and music blur the difference between the 2 groups? Answer: No, the in-person training group did not receive any videos and music. I think there is writing mistake and the manuscript was edited based on this comment. 

RESULTS

1. Think about showing your full statistical workings and outcomes in the tables. Answer: the results were edited (page 16-19). 

2. Clarify in Table 2 (and other tables where needed) whether bracketed numbers are the standard deviation or a percentage of the non-bracketed number directly before. Answer: Based on your comment, the tables were edited.

3. P19 – text underneath table 4 is confusing – please clarify your meaning here – what did the statistical outcomes signify? Answer: this part was edited (page 19)

DISCUSSION

1. The findings are important in that FOC is a crippling psychological condition that can results in unnecessary interventions. Could a larger sample size be used in a future study? Answer: Yes of course, this item was added (highlighted by yellow color at page 22 line 3-5).

2. It might also be mentioned that a mixture of online and face to face classes is more manageable for pregnant women who have other children or who have to travel. 

Based on your comment, this item was added (highlighted by yellow color at page 22 line 20-22).

GENERAL 

Ideally this paper needs to be proofread and edited by a fluent English speaker as there are differences in grammar and turns of phrase that interfere with a smooth reading process. More importantly, the semantics are sometimes lost and your results could be misinterpreted during reading although ultimately they become clear. Answer: the whole of manuscript was edited by a native speaker (Dr. Sally Pezaro as one of the co-authors). 

Some points

Thank you for including your ethics statement: "The present study, while obtaining the code of ethics with the number: IR.IUMS.REC1396.9511373011 from the Research Deputy of XX University of Medical Sciences has been registered in the Iranian clinical trial registry with the code: IRCT201804447070394436N2. A written informed consent was obtained from all the study participants. "Please amend your current ethics statement to include the full name of the ethics committee/institutional review board(s) that approved your specific study. 

Once you have amended this/these statement(s) in the Methods section of the manuscript, please add the same text to the “Ethics Statement” field of the submission form (via “Edit Submission”). Answer: it was edited (page 16). 

If you have any questions or concerns, please do not hesitate to contact me.

Sincerely,

Leila Amiri-Farahani 

Corresponding author

PhD in Reproductive health, Iran University of Medical Sciences, Tehran, Iran 

l.amirifarahani@gmail.com

---

## [Decision Letter · Decision Letter 1]

25 Jul 2022

Comparing the effect of childbirth preparation courses delivered both in-person and via social media on pregnancy experience, fear of childbirth, birth preference and mode of birth in pregnant Iranian women: A quasi-experimental study

PONE-D-21-05419R1

Dear Dr. Amiri-Farahani,

We’re pleased to inform you that your manuscript has been judged scientifically suitable for publication and will be formally accepted for publication once it meets all outstanding technical requirements.

Kind regards,

James Mockridge

Staff Editor

PLOS ONE

Reviewers' comments:

Reviewer's Responses to Questions

**Comments to the Author**

1. If the authors have adequately addressed your comments raised in a previous round of review and you feel that this manuscript is now acceptable for publication, you may indicate that here to bypass the “Comments to the Author” section, enter your conflict of interest statement in the “Confidential to Editor” section, and submit your "Accept" recommendation.

Reviewer #2: (No Response)

Reviewer #3: All comments have been addressed

2. Is the manuscript technically sound, and do the data support the conclusions?

Reviewer #2: (No Response)

Reviewer #3: Yes

3. Has the statistical analysis been performed appropriately and rigorously? 

Reviewer #2: (No Response)

Reviewer #3: Yes

4. Have the authors made all data underlying the findings in their manuscript fully available?

Reviewer #2: (No Response)

Reviewer #3: Yes

5. Is the manuscript presented in an intelligible fashion and written in standard English?

Reviewer #2: (No Response)

Reviewer #3: Yes

6. Review Comments to the Author

Reviewer #2: (No Response)

Reviewer #3: (No Response)

7. PLOS authors have the option to publish the peer review history of their article (what does this mean?). If published, this will include your full peer review and any attached files.

Reviewer #2: No

Reviewer #3: No

---

## [Editor Report · Acceptance letter]

27 Jul 2022

PONE-D-21-05419R1 

 Comparing the effect of childbirth preparation courses delivered both in-person and via social media on pregnancy experience, fear of childbirth, birth preference and mode of birth in pregnant Iranian women: A quasi-experimental study 

Dear Dr. Amiri-Farahani:

I'm pleased to inform you that your manuscript has been deemed suitable for publication in PLOS ONE. Congratulations! Your manuscript is now with our production department. 

Kind regards, 

on behalf of

Dr James Mockridge 

Staff Editor

PLOS ONE